# Molars to Medicine: A Focused Review on the Pre-Clinical Investigation and Treatment of Secondary Degeneration following Spinal Cord Injury Using Dental Stem Cells

**DOI:** 10.3390/cells13100817

**Published:** 2024-05-10

**Authors:** Sandra Jenkner, Jillian Mary Clark, Stan Gronthos, Ryan Louis O’Hare Doig

**Affiliations:** 1School of Biomedicine, Faculty of Health and Medical Sciences, University of Adelaide, North Terrace, Adelaide 5000, Australia; sandra.jenkner@adelaide.edu.au (S.J.); stan.gronthos@adelaide.edu.au (S.G.); 2Neil Sachse Centre for Spinal Cord Research, Lifelong Health Theme, South Australian Health and Medical Research Institute, North Terrace, Adelaide 5000, Australia; jillian.clark@adelaide.edu.au; 3Adelaide Medical School, Faculty of Health and Medical Sciences, University of Adelaide, North Terrace, Adelaide 5000, Australia; 4Mesenchymal Stem Cell Laboratory, Precision Medicine Theme, South Australian Health and Medical Research Institute, North Terrace, Adelaide 5000, Australia

**Keywords:** spinal cord injury, neurotrauma, mesenchymal stem cells, dental stem cells, secondary injury

## Abstract

Spinal cord injury (SCI) can result in the permanent loss of mobility, sensation, and autonomic function. Secondary degeneration after SCI both initiates and propagates a hostile microenvironment that is resistant to natural repair mechanisms. Consequently, exogenous stem cells have been investigated as a potential therapy for repairing and recovering damaged cells after SCI and other CNS disorders. This focused review highlights the contributions of mesenchymal (MSCs) and dental stem cells (DSCs) in attenuating various secondary injury sequelae through paracrine and cell-to-cell communication mechanisms following SCI and other types of neurotrauma. These mechanistic events include vascular dysfunction, oxidative stress, excitotoxicity, apoptosis and cell loss, neuroinflammation, and structural deficits. The review of studies that directly compare MSC and DSC capabilities also reveals the superior capabilities of DSC in reducing the effects of secondary injury and promoting a favorable microenvironment conducive to repair and regeneration. This review concludes with a discussion of the current limitations and proposes improvements in the future assessment of stem cell therapy through the reporting of the effects of DSC viability and DSC efficacy in attenuating secondary damage after SCI.

## 1. Introduction

Spinal cord injury (SCI) is a debilitating condition caused by damage or disease of the spinal cord. It can result in long-term or permanent loss of mobility, sensation, and/or autonomic function due to the impaired conduction of descending and ascending neurotransmission. SCI consequently leads to an increased risk of premature mortality and often generates severe comorbidities, including, but not limited to, chronic neuropathic pain, sexual dysfunction, bowel and bladder dysfunction, immunocompromise, and mental health and well-being disturbances. This causes serious physical, environmental, societal, and economic burdens for patients and their families [1]. 

It is estimated that 900,000 new cases of traumatic SCI occur annually on a global scale [2], with the lifetime economic costs to individuals with SCI estimated between USD 1.2–2.5 million [3]. Importantly, these numbers are expected to rise with improved care and the increased life expectancy of individuals living with a SCI. Despite this high incidence and a wealth of knowledge on the pathophysiology of SCI, there is currently no treatment to halt or reverse the neurological deficits within the spinal cord or prevent the etiology of comorbidities.

## 2. Pathophysiology of the Secondary Injury Cascade following Spinal Cord Injury

The pathophysiology of traumatic SCI impacts the neural structures within the spinal cord as well as each glial population in different ways and involves a complex and unique multicellular response. Similar to other central nervous system (CNS) trauma (e.g., traumatic brain injury (TBI), concussion, and stroke), SCI involves both primary and secondary injury mechanisms. Primary injury refers to the immediate mechanical impact or physical disruption to the neural tissue (reviewed in [4]) that occurs at the time of trauma. While the complete transection of the spinal cord is rare, hemi-sections, partial tearing, or contusion injuries are more common in the clinical setting. Neural tissue surrounding the primary injury area becomes vulnerable to degeneration known as secondary injury, which involves biochemical and cellular cascades of events that exacerbate and propagate the initial injury (Figure 1) and worsen functional outcomes [5]. Secondary events can be temporally divided into acute (<48 h), subacute (48 h to 14 days), intermediate (14 days to 6 months), and chronic (>6 months) phases. However, secondary injury often lasts throughout an individual’s lifetime, preserving a hostile environment that prevents the complete healing or regeneration of the injured area. Given the progressive nature of the secondary injury cascade, the multiple pathological and physiological mechanisms involved provide important therapeutic targets. The following sections provide background information on the biochemical events, cascades, and biological factors observed during secondary injury after SCI that are relevant to the therapeutic actions of stem cells discussed later in this review. 

### 2.1. Vascular Events—Hemorrhage and Edema 

Typically, mechanical impacts to the spinal cord result in immediate disruption and damage to the surrounding microvasculature of the blood–spinal cord barrier (BSCB). Numerous hemorrhages occur within the lesion immediately, with gray matter most affected due to the total cessation of blood perfusion, which can last more than 24 h [6]. Ischemia also arises as a result of vasospasms, endothelial cell swelling, and edema [7]. When compromised by vascular events, the spinal cord becomes highly susceptible to further damage above and below the injury site. However, the extent of these events is dependent on the type and severity of the injury [8]. Within white and gray matter, the leakage of plasma fluid into the meningeal compartments and extracellular space, as well as loss of ionic homeostasis, is observed [7,9,10], including disturbances in intracellular calcium (Ca^2+^) levels that elicit swelling of blood vessels, neurons, and glia [11]. Blood–spinal cord barrier integrity is disrupted within 5 min up to 28 days post-injury [9,12], evidenced by the dissociation of pericytes surrounding microvessels [7] permitting the indiscriminate passage of substances mediated by endothelin-1 and matrix metalloproteinases (MMPs) [13]. Ischemic damage with further cell death is maintained chronically after injury due to systemic hypotension and hypoxic tissue damage [14], in part mediated by the loss of pericyte function [15]. Unfortunately, the re-establishment of normal blood flow to the spinal cord causes reperfusion injury, exposing neural tissue to destructive biochemical factors and sustained influxes of immune cells and inflammatory cytokines in the acute to intermediate phases of injury, triggering further secondary degeneration.

### 2.2. Biochemical Events

#### 2.2.1. Excitotoxicity

Glutamate excitotoxicity is a complex pathological mechanism caused by the excessive or prolonged activation of excitatory receptors via glutamate [16,17], the major excitatory transmitter of the mammalian CNS. After SCI, glutamate is released from damaged neural tissue at neurotoxic levels [18], and insufficient clearance by surviving astrocytes leads to the prolonged excitatory activation of glutaminergic ionotropic *N*-methyl-*D*-aspartate (NMDA) and a-amino-3-hydroxy-5-methyl-4-isoxazolepropionic acid (AMPA) receptors. Glutamate excitotoxicity contributes to large intracellular Ca^2+^ fluxes into neurons and glia [19], resulting in mitochondrial dysfunction, the activation of various enzymes, free radical and nitric oxide formation, and neural and oxidative stress [18,20]. Furthermore, the dysregulation of NMDA and AMPA receptors coupled with the loss of adenosine triphosphate (ATP)-dependent ionic gradient regulation increases the toxic accumulation of sodium and water within axons (for a review, see [21]). This ionic imbalance causes swelling that contributes to further mechanical damage.

#### 2.2.2. Oxidative Stress

Oxidative stress is induced by the overproduction of reactive oxygen species (ROS), such as free oxygen radicals or nitric oxide by-products, which reach damaging levels in the first few hours of injury. The spinal cord is particularly prone to oxidative damage due to limited antioxidant defense capabilities, a large presence of polyunsaturated fatty acid chains sensitive to oxidation [13], and mitochondria shown to be up to 10 times more sensitive to oxidative damage than mitochondria in the brain [22]. SCI patients also exhibit a paucity of plasma antioxidants and, thus, insufficient oxidative balance for up to 12 months after injury [23]. Mitochondrial dysfunction has been demonstrated to be an initial source and target of oxidative stress, caused by excessive intracellular calcium influx, the inability to sequester this calcium, and resultant ROS production [24]. End products of oxidative stress, 4-HNE and 3-NT, cause further excessive damage to mitochondria, resulting in increased permeability, respiratory dysfunction, and mitochondrial death [25]. Increased nitric oxide content and protein oxidation, evidenced through protein carbonyl and 3-NT increases, occur within 24 h of injury [26]. Lipid peroxidation, the oxidative attack of the phospholipid bilayer of cell membranes, occurs later in the injury stage and spreads along the length of axons, causing the disruption of membrane transport, ionic gradient imbalances, oligodendrocyte damage, membrane lysis, and the death of previously unaffected cells [14,27]. Particularly prevalent in SCI, ROS are also formed within hypoxic environments as a result of ischemia from the by-products of hemoglobin breakdown and the phagocytosis of myelin debris by inflammatory cells, as well as during the reperfusion of ischemic tissue [13], with neutrophils and microglia implicated as the main producers of ROS within the damaged spinal cord [28]. 

#### 2.2.3. Apoptosis and Cell Death

Secondary injury cascades in the acute to intermediate phases, particularly excitotoxicity and oxidative stress, culminate in the apoptotic death of neurons and glial cells. Apoptosis begins within hours of the injury, with 90% of neurons lost in the first 8 h within the lesion site [29]. Oligodendrocyte apoptosis can begin as early as 15 min post-injury [29], spreading to otherwise unaffected cells distant from the lesion site and leading to extended demyelination, impaired nerve conduction, and axonal degeneration [30]. The apoptosis of cells in the spinal cord is initiated through multiple pathways. Caspase pathways, the most critical initiators of apoptosis, including caspase-3, caspase-8, and caspase-9, are highly activated within 1 h to 1 day post-injury [30,31]. The Bcl2/Bax pathway, which regulates mitochondrial-mediated apoptosis, is also significantly dysregulated post-injury, with the anti-apoptotic Bcl2 protein being continuously downregulated up to 3 weeks post-injury, and the pro-apoptotic Bax protein elevated in a time-dependent manner [32], leading to a continued cycle of cell death. The various contributions of other forms of programed cell death to the pathophysiology of SCI beyond the scope of this review, such as ferroptosis, pyroptosis, and autophagy, have been reviewed in [33]. 

### 2.3. Inflammatory Events—Neuroinflammation and Immune Cell Influx

The breakdown of the BSCB exacerbates the neuroinflammatory response, orchestrated by residential glial activation and infiltrating leukocytes (neutrophils, monocytes, and lymphocytes). This response occurs both within the acute lesion and distally, and results in the influx of further infiltrating immune cells and the activation of microglia/macrophages. Most prevalently after SCI, macrophages and microglia become activated after exposure to pro-inflammatory mediators, such as interferon-gamma (IFN-γ), tumor necrosis factor-alpha (TNF-α), or lipopolysaccharide (LPS) [34], and exhibit a phenotype that results in pro-inflammatory cytokine secretion, phagocytosis, and collateral damage [35]. Microglia/macrophages and astrocytes secrete high levels of pro-inflammatory interleukin (IL)-12, IL-23, IL-1β, TNF-α, and IL-6, and low levels of anti-inflammatory IL-10, IL-4, and IL-13 [36,37,38], which cause further damage to vulnerable neural tissue. An excessive and dysfunctional increase in pro-inflammatory cytokines, interferons, and prostaglandins in acute SCI contributes to a cyclical influx of inflammatory cells and injury exacerbation at later injury stages. 

Influxes of ROS and protease releasing neutrophils, which appear within the first hour post-injury, peak at 24 h and begin to diminish by 48 h [28], promoting increased vascular permeability and leukocyte influx. B and T lymphocytes infiltrate the cord within the first few hours of injury, declining by 7 days [39]. Microglial macrophages exhibit the greatest activation at 3 and 7 days, whilst monocyte-derived macrophage infiltration is the greatest by 7 days and persists for weeks, months, or indefinitely [28,40]. Reactive astrocytes are most prominent at chronic time points, but begin to display aberrant hypertrophy and proliferation sub-acutely in the first stage of glial scar formation [41]. Interestingly, the inflammatory response within the spinal cord greatly exceeds that of similar traumatic injuries in the brain [42]. 

### 2.4. Structural Events—Axonal and Myelin Changes, Glial Scarring, and Wallerian Degeneration

Structural damage, including the dysfunction and degeneration of neuronal cell bodies, axons, and glia, contributes substantially to the loss of neurological function and poor prognoses following SCI. Significant axonal changes begin within 15 min of injury and evolve over the acute phase. Acute changes include axonal fragmentation, swelling, organelle spillage into the extracellular space, myelin sheath thinning, and calcification [43,44], resulting in the necrotic death of cells and tissue. Cells with greater damage experience accelerated necrotic death, and overall lesion severity dictates the number of lost neurons and glia [45]. The intermediate and chronic phase of injury is marked by the stabilization and maturation of the lesion. This involves the formation of permanent cystic cavities within the lesion epicenter filled with extracellular fluid, connective tissue, and macrophages. The lesion is subsequently surrounded by a glial scar barrier composed of pericytes, fibroblasts [46], reactive astrocytes, microglia, and their secreted products, including chondroitin sulfate proteoglycans (CSPGs) and NG2 proteoglycan [47]. 

Degenerating oligodendrocytes and myelin sheaths release factors, including neurite outgrowth inhibitor A (Nogo-A) [48] and myelin-associated glycoprotein, which activate the Rho-associated protein kinase (ROCK) pathway to initiate growth cone collapse and neurite retraction [49]. Additionally, oligodendrocyte precursor cell depletion contributes to myelination defects or the insufficient remyelination of damaged axons [50]. Myelin proteins from damaged myelin sheaths accumulate, causing auto-immune reactivity mediated by lymphocytes that causes further myelin destruction and tissue dysfunction [51]. Furthermore, damaged axons undergo the slow process of Wallerian degeneration, resulting in subsequent remote tissue damage [52]. The resultant tissue environment is both inhospitable and inhibitory to repair, remyelination, or de novo pathway development. Syrinx formation, which occurs in approximately 30% of patients, leads to the development of large, fluid-filled, and high-pressure cavities with no clear etiology, which can greatly extend the lesion. An ascending lesion can contribute to neurological deterioration, brainstem involvement, or late-onset neuropathic pain [53]. 

## 3. The Benefits of Stem Cells for SCI Therapy 

Numerous therapeutics designed to counter secondary injury pathophysiology or promote spinal cord repair have been studied in various pre-clinical animal models, most commonly in rodents (reviewed in [54]). The most common therapies are pharmacological (drugs or trophic factors) and cell or cell-derived in nature [54,55]. Arguably, the most studied yet controversial pharmacological agent for the management of SCI in the clinical setting is methylprednisolone sodium succinate [56]—a synthetic glucocorticoid receptor agonist that reduces oxidative stress, excitotoxicity, and pro-inflammatory neuronal phagocytosis [57]. Whilst promising results have been demonstrated in pre-clinical studies, the results of recent pharmaceutical clinical trials lack significant outcomes [58]. As reviewed by Zhang et al., 2021, there is still no safe and effective pharmacological or non-pharmacological treatment for the restoration of neurobiological function [59]. The pathophysiology of SCI is mutifaceted and rich with therapeutic targets. However, most of the pre-clinical investigations have been focused on a single component of the complex injury cascade. A multifactorial or multimodal approach to secondary injury may be necessary in order to achieve realistic functional improvements. Additionally, due to the limited growth capacity of the CNS and prohibitive cellular–molecular environment, treatments must be able to create a microenvironment that is conducive to endogenous or exogenous repair, as well as contribute to the inter-dependent biological processes of cellular regeneration and tissue restoration [27]. 

The growing interest in the treatment of SCI using stem cells is attributed to their ability to replenish lost neural and glial cells, as well as foster an environment conducive to endogenous or exogenous repair. Niches harboring tancyte-like cells expressing neural precursor markers within the spinal cord ependymal region surrounding the central canal have been postulated as a potential source of reparative endogenous stem cells [60,61]. However, evidence for the functional complexity of this ependymal niche mainly derives from investigations of mice or rat neonates [62]. Contradictory evidence comes from more recent studies indicating important interspecies differences in spinal cord and stem cell niche anatomy. For example, in comparison to non-primate mammals, the human spinal cord lacks central canal patency and proliferative and regenerative potential of surrounding ependymal cells [63,64]. These findings support the use of exogenous stem cell transplantation as an SCI therapy. Numerous candidate stem cell subsets harvested from different tissue sources are under active evaluation at the pre-clinical and clinical trial phases of translation, including progenitor cells, induced pluripotent stem cells, and glial cells (for an extensive review, see [65]). However, for stem cell transplantation to navigate the translational pathway and be adopted as a clinical therapy, the cell type must be: either available from a stem cell biobank, or be readily and noninvasively available from a viable donor; rapidly and easily expandable with limited ethical considerations; and have an acceptable risk profile. 

### 3.1. Mesenchymal Stem Cells 

Mesenchymal stem cells (MSCs) are one of the most commonly utilized therapeutic stem cell subsets, demonstrating safety and efficacy in clinical trials [66]. These adult multipotent cells are derived from multiple tissue types, including bone marrow and adipose tissue, and characterized by their expression of CD105, CD73, and CD90 and their lack of the expression of hematopoietic markers, including CD45, CD34, CD19, and CD11b [67,68]. MSCs have demonstrated a capacity to differentiate into chondrocytes [69], myofibers [70], and osteoblasts [71]. The differentiation potential of MSCs into neuronal-like cells has previously been demonstrated [72,73] but equally challenged in recent decades [74]. MSCs offer several advantages over engineered stem cell subsets, including simple extraction, minimal ethical considerations, no reprogramming costs, and less epigenetic uncertainty [75,76]. Their limited tissue-specific differentiation capacity, low tumorgenicity [77], immunomodulatory capabilities, and low immunogenicity [78,79] make them a favorable treatment option for CNS injuries [77,80]. 

### 3.2. Dental Stem Cells

Dental stem cells (DSCs) are a group of MSC-like cells (Figure 2) [81] that are gaining traction for their application as a potential neuroprotective and neuro-regenerative tool. First isolated by Gronthos et al. in 2000 [82], DSC populations reside in specialized tissue [83], express MSC markers (including CD90, CD73, CD105, CD44, and STRO-1 [84]), and display multi-differentiation potential, with the capacity to give rise to osteo/odontogenic, adipogenic, and neurogenic cell lineages [85,86].

DSCs exhibit qualities that make them advantageous to the more intensively investigated MSC neuro-regenerative therapies (Figure 2). DSCs originate from the neural crest and express neural markers even before neural differentiation, including microtubule-associated protein-2 (MAP-2), glial fibrillary acidic protein (GFAP), nestin, and β-III tubulin [87], particularly compared to MSC expression profiles [88], which have the potential to aid in neural differentiation [89]. These cells exist at a higher density in dental tissues than stem cells within bone marrow niches, have a higher proliferation rate than other MSC populations including bone marrow MSCs (BMSCs) [82], and have demonstrated encouraging regenerative potential in both peripheral and neural tissue, often with enhanced capabilities for neural regeneration compared to MSCs [90]. Of importance to attempts to design viable SCI therapeutics, DSCs are extracted non-invasively, either from discarded dental tissue or molar tooth removal [91]. The simplicity of donor harvest greatly limits the ethical considerations involved with extraction, which present for alternative stem cell populations [75]. A significant advantage involves autologous engraftment, and aligned to this, a reduced risk of immune reactivity. Furthermore, DSCs lack the expression of the major histocompatibility complex class II receptor (MHC II), preventing antigen recognition by the immune system [92]. As such, xenotransplantation of human DSCs has been investigated in rodent models of SCI with no immunosuppressive regimens [93,94], demonstrating no toxicity or cell rejection. Therefore, the immunomodulatory capacity and low immunogenicity of DSCs also permits allogeneic engraftment without the use of immunosuppressants [80], important in acute cases of neurotrauma for SCI individuals with already immunocompromised immune systems. 

## 4. DSC Modulation of Secondary Cascades after SCI 

Mesenchymal and dental stem cells have been extensively investigated in the field of neuro-regeneration and have demonstrated the ability to modulate the lesion environment after SCI to attenuate secondary injury (Figure 3). Following a PubMed search using the search term strategies shown in Appendix A, the literature to date presents 30 pre-clinical xenogeneic in vivo animal studies of SCI reporting the therapeutic effects of DSCs alone, DSC-conditioned media (CM), DSC exosomes, or DSCs combined with scaffolds, hydrogels, or drug therapy. Most studies delivered an acute intraspinal dose of DSCs (1 × 10^5^–2 × 10^6^) in rat models of spinal compression or contusion. Almost all the literature reported the neuro-recovery benefits of DSC therapy; however, only nine (30%) studies reported on the effects of DSC therapy on three or more of the key secondary injury mechanisms described above. Structural and anti-inflammatory or structural and anti-apoptotic or cell death events were mostly reported, whereas the reporting of the attenuation of biochemical events, such as glutamate excitotoxicity and oxidative stress, was uncommon, highlighting a critical research gap that requires further investigation. Discussed in detailed sections below and summarized in Table 1 are these studies and their specific investigations of DSCs in SCI and their effects on secondary injury. 

Accumulating evidence highlights that the greatest therapeutic benefit following stem cell administration into a neurodegenerative (for a review, see [122]) or injured CNS does not come from the replenishment of neural or glial cell populations, but from the propagation of a supportive environment. Paracrine signaling and cell-to-cell interactions, in which DSCs have demonstrated superior capabilities compared to MSC populations, appear to be key to this success [123,124]. Limited direct comparisons between DSCs and other MSCs in the context of neuro-regeneration and the attenuation of secondary biochemical injury exist. Therefore, although less consequential to the application of regenerative medicine, in vitro studies are included in Table 2 to demonstrate biochemical mechanisms that are superior in DSCs compared to other stems cell types, or vice-versa; however, in vivo investigations are necessary to substantiate the claims of these studies in clinically relevant models. Despite this limitation, a distinction can be delineated, suggesting the stronger beneficial effects of DSCs over MSCs (Table 2). Due to a paucity of literature of relevance to SCI, the remainder of this review also draws on the literature from in vitro studies as well as other CNS neurotrauma, such as TBI and stroke, where relevant. 

### 4.1. Angiogenesis

The ability of a therapy to revascularize ischemic CNS tissue is essential to provide blood supply and initiate repair of the damaged area. In vitro analyses demonstrate the extensive secretion of the angiogenic factor vascular endothelial growth factor (VEGF) [133,134] by DSCs at significantly higher levels than BMSCs [107,130]. DSC-secreted VEGF was found to induce the migration of endothelial cells towards DSCs and increase endothelial tubulogenesis in vitro within 24 h [135,136]. Dental stem cells, particularly those derived from the dental pulp, have also shown powerful angiogenic and vasculogenic potential in vivo. In an in vivo fertilized chick egg assay, DSCs increased blood vessel formation within 3 d [135,136], and increased vascularization and functional blood perfusion in new tissue growth after scaffold implantation in rats [100]. Of note, more prolific vessel formation was observed after DSC transplantation compared to BMSCs in a validated model of stroke [130]. DPSCs and MSCs were observed to localize around blood vessels in vitro [130] and in an in vivo model of SCI [137], respectively, appearing to act as stabilizing structural support cells for pericytes and other cell types involved in de novo angiogenesis. 

Adipose tissue-derived MSCs (AMSCs), exosomes derived from BMSCs [137,138], and DSCs [121] have also been observed to improve the BSCB compromise that accompanies a SCI by supporting the maturation of neovascularization in rats, via a mechanism that is postulated to promote pericyte proliferation. Specifically, DSCs were found to differentiate into pericytes that could regulate vascular function to reduce hypoxia after SCI [121]. In other SCI rodent models, IV-engrafted BMSCs [139] as well as the intra-spinal engraftment of DSCs [115,116] reduced intraspinal hemorrhage, although further analysis is needed to elucidate the mechanistic basis of this attenuated hemorrhage. The intra-spinal engraftment of DSC hydrogels also stimulated angiogenesis, blood perfusion, and blood vessel organization within the intraspinal lesion epicenter in mouse and rat models [106,117,120]. After engraftment into a complete transection SCI rat model, angiogenesis and increased blood vessel density within sensory tract areas (measured by CD31 staining) were stimulated by DSC-loaded scaffolds, promoting sensory fiber regeneration and improving sensory function [100]. Of importance, studies comparing the effects of BMSCs with DSCs within a rat cerebral ischemic injury model showed increased blood vessel formation in the groups treated with DSCs, implying a mechanism of action of DSCs that is not common to MSCs [129]. Interestingly, a recent pre-clinical study reported that, whereas AMSCs secreted large amounts of various pro-angiogenic factors in vitro, including VEGF, these pro-angiogenic profiles where not maintained under in vivo conditions (PDGF-AA, endothelin-1, TIMP-1, and Serpin-E1) [137]. These data highlight the need for further in-depth histological analyses. 

### 4.2. Anti-Excitotoxic Effects

In in vitro neuronal cultures, MSCs and DSCs have been demonstrated to confer significant protection against glutamate- and NMDA-mediated excitotoxicity. DSCs were able to increase the viability and survival rate of neurons cultured under excitotoxic conditions [124,140], and AMSCs and BMSCs restored mitochondrial function, ATP production, and NAD^+^/NADH mitochondrial respiration substrates as well as inhibited NMDA receptor subunit expression in neurons [141,142]. Although not extensively investigated in vivo, MSCs and DSCs are also able to reduce excitotoxicity in animal models of SCI. Watanabe et al. (2015) reported that BMSCs reduce the expression of several markers (e.g., PKC-y and p-CREB) implicated in the pathophysiology of glutamate-induced neuronal hyperexcitability and neuropathic pain in spinal neurons in mice [143], while Nishida et al. (2020) demonstrated that human umbilical cord MSCs (UMSC) protect neurons and restore function in a rat model of glutamate-induced cytotoxicity and spinal cord damage [144]. Likewise, the intra-spinal engraftment of SHED was demonstrated to reduce the overexpression of glutamate-induced neuronal nitric oxide synthase (nNOS), as well as the excitatory amino acid transporter 3 (EAAT3) to limit glutamate-mediated cytotoxicity after SCI in rats [110], and Ying et al. (2023) demonstrated a reduction in over-active glutaminergic synapses and increased GABAergic inhibitory synapses after DPSC engraftment in mice after SCI [117]. However, overall analyses into the anti-excitotoxic effects of DSCs are scarce and require further in-depth investigation.

### 4.3. Anti-Oxidative Effects

In culture with neurons, DSC-CM or DSCs have exhibited the ability to reduce DNA damage, ROS, and NO after oxidative stress [115,145,146,147], and survive hydrogen peroxide-induced oxidative stress with unperturbed growth factor expression [108]. Similarly, AMSC [148] and DSC [149,150] treatment improved neuronal stem cell viability and prevented apoptosis under cytotoxic oxidative conditions in vivo. Interestingly, the findings of Song et al. (2015) demonstrate the superior ability of DPSCs to reduce ischemia-induced astrocyte death in culture when compared to BMSCs [132]. In SCI animal models of stem cell engraftment, DSCs were able to reduce ROS production to counteract ROS-mediated neuroinflammation [104], reduced iNOS levels [103,107] involved in the overproduction of NO, and limited lipid peroxidation (4-HNE staining) while increasing the expression of the GPX4 anti-oxidant [117]. In animal models of other CNS injuries, DSC transplantation was shown to reduce the production of the oxidative stress markers 4-HNE [151] and 3-NT to a greater extent than BMSCs [124] and ROS [152]. A canine model of SCI likewise demonstrated the significant attenuation of 4-HNE and protein carbonyl associated lipid peroxidation following intravenous MSC infusion [153]. As noted for investigations into the anti-excitotoxic effect of stem cells, further characterization is required to deduce the specific antioxidative mechanisms of DSCs and elucidate whether they directly or indirectly affect cell function after SCI. 

### 4.4. Neuroimmunomodulation

Perhaps the most extensively studied and potent capability of mesenchymal and dental stem cells in the context of SCI is their innate ability to ameliorate the harsh and refractory pro-inflammatory cascade. Attenuating pro-inflammation acutely to limit inflammatory dysfunction in the later stages of injury is a necessary feature for all SCI therapeutics. In in vivo models of SCI, BMSCs [154,155] and DSCs polarized macrophages into anti-inflammatory phenotypes by increasing the expression of Arg-1, CD206, and IL-10 in macrophages/resident microglia, with DSCs showing superior capabilities to achieve this when compared to BMSCs [107]. Furthermore, the in vivo engraftment of DSCs reduced the expression of pro-inflammatory iNOS, IL-6, CD16/32, and IL-1β [102] within the spinal cord lesion. Other studies showed that MSCs and DSCs alter macrophage polarization into anti-inflammatory phenotypes within 12 h of SCI engraftment [107], and can maintain an anti-inflammatory environment for up to 10 weeks [156]. Additionally, DSCs influence the inflammatory secretome that accompanies SCI, decreasing the production of the pro-inflammatory cytokines IFN-γ, TNF-α, IL-2, IL-17, IL-6, and IL-1β, normally expressed and maintained at high concentrations following injury, and concurrently increasing the secretion of the anti-inflammatory cytokines IL-4, IL-13, TGF-α, and IL-10 [96,102,107,157]. BMSCs, AMSCs [158,159], and DSCs [105] also modulate inflammasome complexes, including a reduction in NLRP3 expression and associated NF-κB-, IL-1β-, and IL-18-mediated inflammation in animal models of SCI. Importantly, DSCs conduct paracrine immunomodulation through the release of anti-inflammatory cytokines both in vitro [87,157] and in vivo, as shown for SCI models [107], and thus do not rely on cell–cell contact to induce the anti-inflammatory phenotypes of macrophages/microglia. Indeed, Matsubara et al. (2015) demonstrated the strong anti-inflammatory induction of cells in the SCI lesion following the engraftment of conditioned medium derived from SHED alone [107], while other authors found that intravenously injected BMSCs exert paracrine immunomodulatory and trophic effects upon the injured spinal cord [160]. This discovery was found despite no cells appearing to migrate to the spinal cord, which instead settle in the lungs, as revealed by bioluminescence imaging and spectrophotometric quantitation [7,161]. Although stem cell treatment has been discordantly demonstrated to both enrich [154,155] and deplete [162,163,164,165] macrophage/microglia populations acutely after SCI, stem cells ultimately generate anti-inflammatory and reparative phenotypes of immune cells. Of note, DSCs were also demonstrated to reduce microglial pyroptosis and reduce astrocytic glial scar formation in vivo [105,109,110,113]. The immunomodulatory influence of DSCs exerted upon lymphocytes was demonstrated in various in vitro models, including a reduction in the activation and proliferation of NK cells [166] and a reduction in the activation and migration of B cells [167,168], while the corresponding ratio of anti-inflammatory Treg cells was increased [128]. Evidence for the in vivo modulatory effects of DSCs on leukocytes following SCI indicates their ability to suppress T-cell infiltration [110], but a further mechanistic investigation is required.

### 4.5. Anti-Apoptotic Effects

Preventing or inhibiting the cascade of ongoing cell death initiated during the acute response to SCI is an important therapeutic goal. The engraftment of MSCs [162,169,170,171] and DSCs [87,106,110,112,119] has been shown to reduce the number of apoptotic neurons and glia following SCI. A possible mechanism by which stem cells exert these anti-apoptotic effects is through their inhibition of the expression of apoptosis initiating factors such as caspase-3, shown to be significantly reduced within the SCI lesion after DPSC [106] or SHED treatment [110]. Increased and decreased anti-apoptotic Bcl2 and pro-apoptotic Bax expression, respectively, were also detected in vitro [148] and in vivo in DSC-treated animal models of SCI [106].

Matsubara et al. (2015) reported the greater expression of anti-apoptotic and neuroprotective factors in SHED-CM than BMSCs, including nidogen-1, insulin, and NCAM-1 [107]. Neurotrophins play an active role in inducing inhibitors of caspase-3 [172]. Therefore, the extensive expression of neurotrophins (including BDNF, GDNF, NT-3, NGF, IGF1, and CNTF) by BMSCs [165], and superior expression by DSCs [87,90,124], may explain the mechanism of action by which stem cells reduce apoptosis [169]. Evidence is also accumulating on the effects of the stem cell regulation of autophagy following SCI, with animal models showing a promotional effect of BMSCs on autophagy through increased autophagy-related proteins beclin-1 and light chain3-II in neurons, resulting in attenuated apoptosis and improved recovery [158,173]. Overall, the literature supports the conclusion that both MSCs and DSCs can promote neuroprotection and the survival of neural and glial cells within the hostile secondary injury environment. 

### 4.6. Tissue Preservation and Regeneration

Attenuating the secondary injury cascades described above culminates in the preservation of the gross lesion architecture and neuronal and glial connectivity. This includes decreasing the cystic cavity and lesion size and limiting glial scarring through CSPG inhibition as demonstrated by DSCs [107,111]. Improvement in the general organization of neuronal and glial structures within the lesion penumbra is also commonly demonstrated by both MSCs [174] and DSCs in vivo [97,104,106].

Improving the preservation of myelin after injury is a vital therapeutic target. MSCs improve the preservation of myelin sheaths within the lesion center as well as caudally, and increase the thickness of preserved myelin sheaths [154,156,169,174], while DSCs and DSC-CM similarly preserve white matter, increase synapse preservation, and limit oligodendrocyte cell loss after SCI [97,99]. Furthermore, DSCs have exhibited a strong capacity to promote neuritogenesis and filipodial migration towards damaged neurons in vitro [123,131], which has been substantiated in in vivo SCI models demonstrating an increase in axonal regeneration (increased acetylated tubulin, neurofilament, and neural regeneration marker GAP-43 staining) and axon preservation, neurite extension, and sprouting of serotonergic, corticospinal tract, and sensory fibers [87,95,105,106,107,108,109,120]. 

Following in vivo SCI engraftment, the literature has consistently shown that DSCs readily differentiate into GFAP, S-100 and APC/MBP-expressing astrocytes [97,102], Schwann cells [97], and oligodendrocytes [87], and importantly, induce neural progenitor cell proliferation [94,115,118]. While DSCs have also been demonstrated to differentiate into neuronal-like cells in vitro [88] and in vivo after SCI through neuronal marker staining [101,102,116], functional analysis of differentiated neurons is less frequently conducted to substantiate these findings. Nevertheless, as discussed in this review, the therapeutic efficacy of stem cells has been demonstrated to rely less on direct stem cell replacement, and more importantly on the protective and supportive influence of stem cells on resident and infiltrating cells and the tissue microenvironment. 

## 5. Future Perspectives for Improving DSC Therapy Translation

### 5.1. Improving DSC Viability 

Outside of the obvious barriers that inhibit stem cell activity in the lesion site after SCI discussed above, the translation of pre-clinical findings into human clinical trials is limited due to the low survival rate of engrafted stem cells. Most studies report either no surviving cells or survival of only <1–2% after 5–10 weeks [175,176], while many studies do not measure cell survival or migration in the first instance. Low cell survival has been postulated to be caused by persistent and damaging pro-inflammatory signaling [177,178], which sustains a harsh microenvironment at all injury stages, likely preventing immunomodulatory and reparative stem cell activity from occurring in time to limit stem cell death. Indeed, stem cells begin to die at exponential rates within hours of engraftment, [143], which has been demonstrated to negatively impact stem cell efficacy post-engraftment in neurotrauma [177,179,180], with increasing stem cell survival rate and tissue sparing strongly correlating in pre-clinical models [181]. A mechanistic understanding of the causes of stem cell death within SCI secondary injury microenvironments is lacking, which could be key to developing the timeframes and techniques aimed at ameliorating stem cell death and improving functional efficacy following engraftment. Importantly, many investigations into the protective functions of stem cells are conducted in vitro prior to their engraftment (e.g., neurotrophin secretion). A greater understanding of how the multicellular response to neurotrauma impacts these functions will be required to enhance translation in human trials. The recent interest in the therapeutic and immunomodulatory effects of apoptotic stem cells (reviewed in [182]) falls outside the scope of this review, but further highlights the need for an increased understanding into stem cell functions in various microenvironments. 

### 5.2. Optimizing the Delivery of DSC Therapy 

The intensive investigation of stem cell pharmacokinetics is another facet of attempts to achieve therapeutic efficacy. Optimizing the delivery route, timing, and dosage of stem cell treatments to improve stem cell survival and treatment efficacy remains an active goal. While no clinical trial has yet investigated the safety or efficacy of DSC engraftment protocols in participants with a primary diagnosis of spinal cord injury, various MSC phase I and phase II trials have been undertaken (reviewed in [183]). Three main routes for cell delivery are utilized in pre-clinical animal models of SCI: intraparenchymal administration via injection directly into the spinal cord tissue; intrathecal administration via lumbar puncture into the fluid-filled space surrounding the spinal cord; and intravenous administration. Xenotransplantation data for intravenous administration show that only approximately 1–2% of circulating human MSCs engraft into the hemisected spinal cord of the rat, yielding less tissue sparing and inducing more marked immunogenicity than lumbar puncture or intraspinal injection [184]. In comparison, intrathecal administration avoids the systemic circulation; however, only an approximate 8–9% increase in the number of engrafted cells is observed before further viability loss [185]. Intraparenchymal injection is more commonly utilized due to a greater control of cell localization, but cell survival remains low, likely due to immediate exposure to the lesion microenvironment [108,143]. This stereotaxic approach also poses risks to the remnant spinal tissue. Studies investigating the various timings of stem cell administration report better stem cell viability when engrafted at the 3–7 days post-injury. Engraftment occurring at chronic phases of injury also yields only weak therapeutic efficacy [143]. Importantly, no study to date has directly compared the effects of acute versus delayed engraftment on DSC viability. 

Dosage is another key factor of translatability, with high inter-study variability observed within and between human and rodent investigations (reviewed in [186,187], respectively) with a higher median dosage routinely administered in animal studies (4.2 × 10^6^/kg) than in clinical trials (1–3 × 10^6^/kg). A recent meta-analysis found that an increased dose of intra-parenchymal transplanted stem cells (≥1 × 10^6^ at) in the subacute phase of pre-clinical SCI (3–14 days post-SCI) had better therapeutic effects than other protocols [188]. Despite this, the majority of the DSC literature to date reports the delivery of sub-optimal doses (≤1 × 10^6^) of DSCs immediately after surgically induced SCI (≤1 h post-SCI) (Table 1). These variables, coupled with the heterogeneity and scarcity of reported outcomes as presented in the current review, highlight future challenges in progressing the translational potential of DSC therapy to clinical trials. 

### 5.3. Increasing Measurable Outcomes of DSC Therapy

The Basso–Beattie–Bresnahan (BBB) locomotor rating scale in rats [189] and the Basso Mouse Scale (BMS) in mice [190] are the accepted measures of motor improvement. A meta-analysis of MSC data obtained from models of SCI in the rat indicated an overall BBB improvement of 3.9 versus the controls across 83 extracted studies [187]. However, the non-linear nature of motor changes in the BBB scale and an unclear correlation to human function presents difficulties in interpreting relevance and clinical significance. Uncertainty about the clinical effectiveness of stem cell therapy is, in part, attributed to the disappointing lack of motor or sensory improvements and the overall achievement of the desired therapeutic effects. As highlighted in this review, accumulating evidence for the attenuation of secondary injury cascades in a biological response to the engraftment of exogenous stem cells, or stem cell-derived secreted factors, may offer an important adjunctive measure of stem cell effectiveness. However, the in-depth characterization and reporting of standardized biomarkers of secondary injury would be necessary in future investigations. In agreement with Shang et al. (2022), the quality of the pre-clinical literature describing stem cell delivery mode and the dosage and timing of stem cell administration represents a barrier for the field [188]. It is equally important to consider and promote the reporting of negative pre-clinical data to further solidify our understanding of the various treatment variables that are, and are not, favorable to treatment administration and limit study duplication as well as animal and resource wastage. Nevertheless, as discussed in this review, the therapeutic efficacy of stem cell therapy is the subject of active investigation and refinement. Overall, our understanding of the true potential of DSCs across the various degrees of pathological and treatment heterogeneity may be improved by the systematic reporting of a range of outcome measures, within a consensus guideline or framework. 

## 6. Conclusions

Stem cell therapy is a promising strategy for the preservation or restoration of the structure and function of the brain and spinal cord. This review discusses the mechanisms of action of dental and mesenchymal stem cells within the CNS microenvironment during secondary degeneration and constructs a translational framework of stem cell therapies of relevance to spinal cord injury. Evidence for paracrine and cell-to-cell modulation of a range of vascular and biochemical events, inflammatory and CNS cells, their signaling pathways, and secretome is considered. We would propose that the multifunctional properties of stem cells, DSCs in particular, have a multifactorial level of control on infiltrating and resident cells and the inflammatory microenvironment that is independent of their multipotent differentiation potential. 

We encourage investigations of DSCs and other stem cell therapies with increased reporting on stem cell viability and the effects (or lack thereof) on the spectrum of secondary injury mechanisms following SCI. This is vital to not only elucidate the mechanisms by which stem cells survive within and repair the harsh cytotoxic microenvironments of the injured spinal cord, but understand which variables may impact clinical translation, efficacy, and ultimately, therapeutic success. 

## Figures and Tables

**Figure 1 cells-13-00817-f001:**
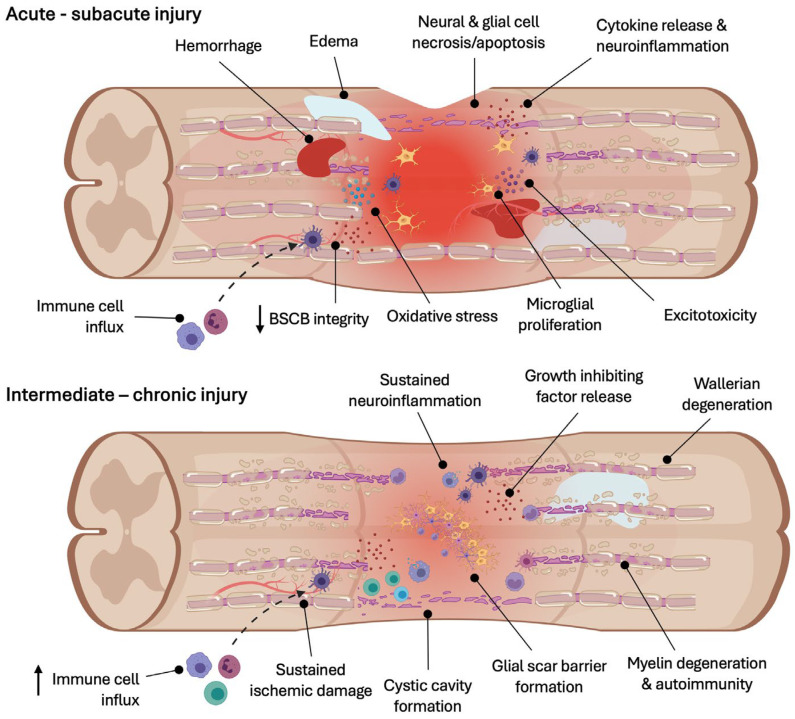
The secondary injury cascade following spinal cord injury involves acute to chronic sequalae that exacerbate the initial mechanical damage to the spinal cord.

**Figure 2 cells-13-00817-f002:**
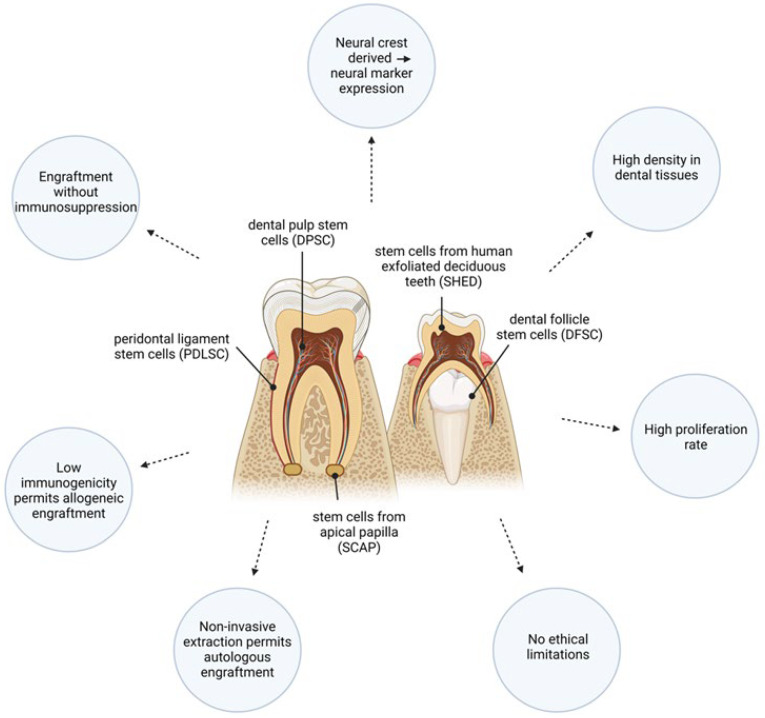
Dental stem cells are a group of mesenchymal-like neural crest derived cells that exhibit favorable qualities for neuro-regenerative therapies.

**Figure 3 cells-13-00817-f003:**
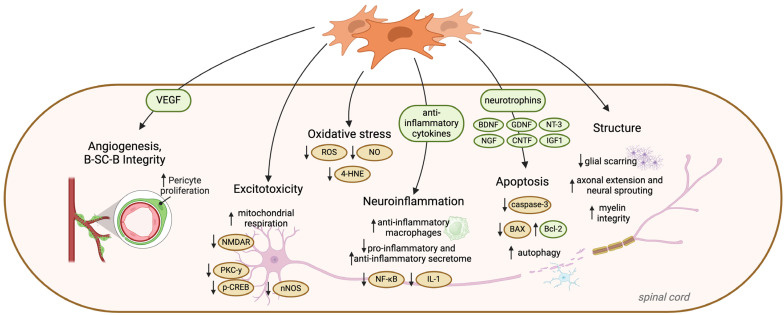
Dental stem cells and mesenchymal stem cells modulate the secondary injury microenvironment after spinal cord injury, attenuating numerous secondary cellular and biochemical cascades to improve functional outcomes after injury.

**Table 1 cells-13-00817-t001:** The literature investigating DSC use in pre-clinical in vivo SCI models. The secondary injury target of each investigation is highlighted, along with specific and significant DSC activity and neurological benefit(s). ✓^V^ indicates investigations also conducted in vitro.

Reference	DSC Type and Groups	DSC Dose	Delivery Method	Injury Model	Vascular Events	Apoptosis and Cell Loss	Biochemical Events	Inflammatory Events	Structural Events	Significant DSC Activity	Functional Benefit(s)
[95]	HeP-hDPSCs, HeP-bFGF-hDPSCs	1 × 10^6^	Intra-spinal—immediate	Mice: Thoracic compression				✓	✓	HeP + bFGF + DPSCs reduced pro-inflammatory factors (decreased in IL-6 and TNF-α); modulation of NF-κB; neuroprotection; promoted neurite and improved cell sprouting; increases in MAP-2 and Ace-Tubulin; nerve repair	Not mentioned
[96]	PF-OMSF/hDPSCs, PF-OMSF@JK2/hDPSCs	1 × 10^6^	Intra-spinal—immediate	Rats: Thoracic compression				✓	✓	Reduced pro-inflammatory factors (decreased in IL-6 and TNF-α); modulation of NF-κB; neuroprotection; promoted neurite and improved cell sprouting; increases in MAP-2 and Ace-Tubulin	Not mentioned
[97]	hDPSCs	8 × 10^5^	Intra-spinal—7d or 28d post-SCI	Mice: Thoracic compression		✓			✓	Increased white matter sparing; increased neurotrophic factor expression, more so in 7 dpi engraftment group; improved tissue preservation	Improved motor function (BMS) in both groups compared to the media control
[98]	SHED-CM, SHED-CM + Col	3 μL SHED-CM	Intra-spinal—immediate	Rats: Thoracic compression						Not mentioned	SHED-CM alone did not have any effects; SHED-CM delivered with a collagen scaffold demonstrated locomotor, motor, sensory, and sensory–motor improvements
[99]	SHED-CM, SHED-CM + Col	3 μL SHED-CM	Intra-spinal—immediate	Rats: Thoracic compression		✓			✓	No effect of SHED-CM alone; SHED-CM delivered with collagen scaffold preserved gray and white matter, reduced lesion volume, limited neuronal cell loss, and limited oligodendrocyte cell loss	Not mentioned
[100]	hDPSCs + scaffold, hDPSCs + scaffold + HAMECs (prevascularized)	0.45 × 10^6^	Intra-spinal—immediate	Rats: Thoracic transection	✓^V^ ✓				✓^V^ ✓	In vitro angiogenesis and neurogenesis; prevascularized DPSC scaffolds promote axon preservation (B3-tub), myelin deposition (MBP expression), and vessel formation and structure (CD31 expression, vessel volume, and vessel density); partially restored spinal cord microstructure	Prevascularized DPSC scaffolds only improved sensory recovery and small improvement in motor recovery
[101]	hDPSCs + PRP, hDPSCs	2.5 × 10^5^	Intra-spinal—3d post-SCI	Rats: Thoracic contusion		✓			✓	DPSCs reduced syrinx formation, apoptosis (TUNEL Assay); DPSCs survival up to 4 weeks and differentiation into neurons (GFAP and NeuN staining)	Greatest motor function (BBB) improvement in hDPSC + PRP group (no comparisons)
[102]	hDPSCs	3 × 10^5^	Intra-spinal—immediate	Rats: Thoracic contusion				✓		Neural and glial cell differentiation (co-expression of GFAP, NF, nestin, BNDF, and vimentin); reduced pro-inflammatory factor expression (IL-1β, MPO, MIP2, and IL-6) and increased anti-inflammatory expression (IL-1ra and EP3)	Improved motor function (BBB) in DPSC-treated group
[103]	hSCAP, hSCAP + PAMs, hSCAP + BDNF-PAMs	2 × 10^5^	Intra-spinal—immediate	Rats: Thoracic contusion				✓	✓	Reduced CD68+ inflammation; reduced iNOS staining; GAP-43 and βIII-Tubulin axon growth; serotenergic fiber growth	Greatest motor function (BBB) improvement in SCAP + BDNF-PAMs compared to both SCAP and the vehicle control
[104]	hDPSC-derived exosomes	N/A	Tail vein—30 min post-SCI	Mice: Thoracic contusion			✓	✓^V^ ✓	✓	In vitro and in vivo LPS-induced ROS reduction, reduced M1 macrophage polarization and reduced P-ERK/ERK levels; in vivo reduction in M1 macrophage number; slight neuronal preservation (NF200 and NeuN staining) and histological reductions in structural damage	Improved motor function (BMS) in the exosome-treated group
[105]	hDPSCs	200μL DPSC-CM	Intraperitoneal—daily for 3d post-surgery	Rats: Thoracic contusion		✓		✓^V^ ✓	✓	In vitro reduction in LPS-induced NLRP3, CASPASE-1, IL-1β, and IL-18; reduced lesion volume; improved motor-evoked potentials and somatosensory-evoked potentials in anterior fontanelle and hind limb skeletal muscles, respectively; reduced NLRP3, IL-1β, and IL-18 in vivo; reduced microglial pyroptosis; enhanced neural repair (NF200, Tuj1, and MBP staining); reduced glial scarring (GFAP staining)	Improved motor function (BBB, inclined plane test) compared to the untreated group
[106]	HeP-hDPSCs, HeP-bFGF-hDSPCs	10 μL of hydrogel w/ or w/o cells (no cell dose provided)	Intra-spinal—immediate	Rats: Thoracic compression		✓			✓	Reduced apoptotic factor expression (Bax and Caspase-3) and increased anti-apoptotic factor expression (Bcl2) in HeP-bFGF-DSPC group; increased neurogenesis (GAP43) and myelination (MBP); increased tissue and ventral motor neuron preservation	Greatest motor function (BBB, inclined plane test) and sensory function (Reuters test) improvements in HeP-bFGF-DSPC and HP-DPSC groups
[107]	SHED, SHED-CM	1 × 10^6^ + 1 × 10^5^	Intra-spinal fibrin glue + intrathecal pump CM—immediate	Rats: Thoracic contusion				✓^V^ ✓	✓	SHED and SHED-CM reduced tissue loss and spared serotonergic fibers and lesion size; SHED-CM suppressed pro-inflammatory mediators for 1 wk after injury (IL-1β and TNFα), increased expression of anti-inflammatory IL-10, TGF-β1, VEGF, CD206, and Arg-1; increased M2 macrophage phenotypes; in vitro M2 macrophage phenotype induction	SHED improved motor function (BBB) compared to the PBS control; SHED-CM improved motor function (BBB) compared to the DMEM control and BMSC-CM
[108]	hDPSCs + FGF2, hDPSCs	1 × 10^6^	Intra-spinal—immediate	Rats: Thoracic transection	✓				✓	DPSCs and DPSC-FGF2 promoted axon regeneration (GAP-43 staining), DPSC-FGF2 more so; DPSC-FGF2 increased VEGF mRNA expression	FGF2-pretreated DPSCs significantly improved motor function (BBB) compared to the vehicle control and DPSC-only treated groups
[94]	SHED	3 × 10^5^	Intra-spinal—1h post-SCI	Rats: Thoracic contusion					✓	SHED increased neural progenitors (vimentin); SHED reduced astrocytic hypertrophy (GFAP)	Improved motor function (BBB) compared to the untreated group
[109]	SHED, SHED + TT	3 × 10^5^	Intra-spinal—1h post-SCI	Rats: Thoracic contusion				✓	✓	SHED treatment only reduced cystic cavity areas and glial–scar barrier (GFAP) caudally; SHED only increased myelin (MBP) and axonal preservation (NF-M); SHEDs only reduced intra-spinal TNFα levels (ELISA)	SHED improved motor function (BBB) compared to the untreated group
[110]	SHED	3 × 10^5^	Intra-spinal—1h post-SCI	Rats: Thoracic contusion		✓	✓	✓	✓	SHED treatment reduced cystic cavity areas caudally and in the lesion epicenter; motor neuron preservation and reduction in neural apoptosis; reduced T-cell infiltration and TNFα levels; reduced excitotoxic EAAT3 expression	Improved motor function (BBB) compared to the untreated group
[111]	SHED	2 × 10^5^	Intra-spinal—immediate	Rats: Thoracic compression					✓	SHED reduced p-STAT3, GFAP expression; reduced CSPG	Improved motor function (BBB, inclined plane test) compared to the untreated group
[112]	Rat dental pulp	N/A	Intra-spinal—immediate	Rats: Lumbar hemisection		✓				Increased motor neuron survival	None mentioned
[113]	hDPSCs (monolayer-grown), hDPSCs (spheroid-grown)	3 × 10^5^	Intra-spinal fibrin glue—immediate	Rats: Lumbar L4-6 spinal root avulsion		✓		✓	✓	Increased motor neuron survival; reduced astrocyte proliferation (GFAP); reduced microglial proliferation (IBA1); preservation of neural circuitry (synatophysin); mixed inflammatory signaling changes	Monolayer DPSCs improved motor function (peroneal nerve functional recovery, base of support hind paws, max contact area, and step sequence regularity)
[87]	hDPSCs, SHED	1 × 10^6^ + 1 × 10^5^	Intra-spinal fibrin glue—immediate	Rats: Thoracic transection		✓			✓	SHED regenerated transected corticospinal tract and seritonergic axons (DPSC not measured); SHED inhibited Rho GTPase growth inhibitor (DPSCs not measured); SHED preserved myelin sheath (Fluoromyelin and MBP) and differentiated into oligodendrocytes (DPSCs not measured); SHED reduced apoptosis of neural cells (TUNEL assay)	SHED and DPSC groups improved motor function (BBB) compared to the untreated group
[114]	SHED, iSHED	0.5 × 10^6^	Intra-spinal—7d post-SCI	Rats: Thoracic contusion						SHED demonstrated greater affinity for astrocytic differentiation (GFAP); iSHEDs demonstrated greater affinity for oligodendral and neural differentiation (MBP and NG2)	SHED and iSHED improved motor function (BBB), more significant in the iSHED group
[115]	hDPSCs, DPSC-OIC	4 × 10^5^	Intra-spinal—immediate	Mice: Thoracic contusion	✓	✓	✓^V^		✓^V^ ✓	In vitro DPSC supernatant promoted HT-22 cell line axonal length; in vitro DPSC supernatant protected HT-22 cells from H2O2 oxidative stress-induced apoptosis; DPSC-OIC reduced hemorrhage and edema (MR imaging); DPSCs and DPSC-OIC reduced general spinal cord apoptosis (Caspase-3) and increased general cell proliferation (Ki-67); DPSCs and DPSC-OIC increased neural progenitor marker expression (Nestin) and DPSC-OIC increased progenitor marker expression (Sox2); DPSCs and DPSC-OIC reduced axon inhibitory factor NG2 and increased axon growth promoting factor fironectin	DPSCs and DPSC-OIC groups improved motor function (BMS), DPSC-OIC significantly more than DPSCs only at 28d post-SCI
[116]	hDFSCs, hSCAP, hDPSCs	2.5 × 10^5^	Intra-spinal—immediate	Rats: Thoracic transection	✓			✓^V^ ✓	✓	In vitro inhibition of general PBMC proliferation by all stem cells; promoted spinal tissue structure and neuron preservation; reduced IL-1β, RhoA, and ARHGAP growth inhibitory factors, and SUR1 necrosis and hemorrhage by all stem cells; neuronal and oligodendral differentiation (NeuN and MBP staining)	All stem cell groups improved motor function (BBB) compared to the untreated group
[117]	hDPSCs + TPA@laponite shear-thinning hydrogel	Not provided	10 μL hydrogel intra-spinal—immediate	Mice: Thoracic contusion	✓	✓	✓		✓	Reduced lipid peroxidation (4HNE staining); increased neuronal survival closer to injury site (NeuN staining); reduced oxidation promotor expression (NOX2, GPX4, and xCT); preserved tissue integrity; reduction in ferroptosis markers; reduced fibrous blood vessel scarring and improved blood vessel organization; improved axonal regeneration (NF200 staining); regulation of excitotoxicity by reduction in Glutaminergic synapses and increase in GABAergic synapses	Improved motor function (BMS, gait mark analysis, and EMG recordings) compared to the hydrogel only and untreated groups
[118]	hSCAP + ECM gel + Scramsh, hSCAP + ECM gel + MLL1sh	2 × 10^6^	Intra-spinal—immediate	Rats: Thoracic hemisection					✓	MLL1 knockdown in SCAP reduced lesion cavities and scars than SCAP + scramsh group; increased neural progenitors (Nestin staining); increased axonal regeneration (NEFM staining);	MLL1 knockdown in SCAP promoted functional recovery (BBB)
[119]	hDPSCs, hDPSC + chitosan scaffold	2.5 × 10^5^	Intra-spinal—7d post-SCI	Rats: Thoracic contusion		✓			✓	Reduced tissue loss, apoptotic cells and axon degradation (H&E staining); reduced general apoptosis (caspase-3 expression and TUNEL staining)	DPSCs and DPSC + Chitosan scaffold groups improved motor function (BBB), more so in DPSC + Chitosan scaffold group
[120]	hDPSCs + GelMA hydrogel, DPSC + ZIF-8 + GelMA hydrogel	0.5 × 10^6^	10 μL hydrogel intra-spinal—24 h post-SCI	Rats: Thoracic compression	✓	✓			✓	Improved tissue integrity; increased neural and blood vessel regeneration (βIII-tubulin and VEGF-α); restoration of spinal zinc levels; reduced general apoptosis (TUNEL staining)	DPSCs and DPSC + ZIF8 improved motor function (BBB, inclined plan test), DPSC + ZIF8 more so, compared to the untreated group
[121]	hDPSCs, AAV-5HRE-bFGF-DPSCs	5 × 10^5^	Intra-spinal—7d post-SCI	Rats: Thoracic contusion	✓		✓		✓	Differentiation into pericytes, secretion of bFGF, and promotion of pericyte adhesion to vascular endothelial cells to regulate vascular diameter and reduce hypoxia; increased neuron survival and axon regeneration (NeuN and GAP43 staining); inhibited autophagy; reduced astrocytic scar (GFAP and laminin staining)	DPSCs and AAV-5HRE-bFGF-DPSCs improved motor function (BBB, inclined plane test), more so in AAV-5HRE-bFGF-DPSCs group, compared to the untreated group

SHED: stem cells from human exfoliated deciduous teeth; hDPSCs: human dental pulp stem cells; hDFSCs: human dental follicle stem cells; hSCAP: stem cells from apical papilla; CM: conditioned media; BBB: Basso–Beattie–Bresnahan locomotor rating scale; BMS: Basso mouse scale; PBS: phosphate-buffered saline; HeP: heparin hydrogel; bFGF: basic fibroblast growth factor; PF-OMSF: Octyltriethoxysilane functionalized mesoporous silica (MSN) modified with PF-127 hydrogel; PF-OMSF@JK2: hydrogen sulfide gas donor JK2-loaded Octyltriethoxysilane-functionalized MSN modified with PF-127 hydrogel; Col: collagen hydrogel; HAMECs: human adipose microvascular endothelial cells; PRP: platelet-rich plasma; PAMs: pharmacologically active microcarriers; BDNF-PAMs: pharmacologically active microcarriers releasing brain-derived neurotrophic factor (BDNF); FGF2: fibroblast growth factor-2; TT: treadmill training; iSHED: neural induced SHED; OIC: Adenovirus overexpressing osteopontin, insulin-like growth factor 1, and ciliary-derived neurotrophic factor; TPA@laponite shear-thinning hydrogel: TPA(N1-(4-bor- onobenzyl)-N3-(4-boronophenyl)-N1, N1, N3, N3-tetramethylpropane-1, 3-diaminium) ROS scavenger with laponite nanoparticle hydrogel; Scramsh: scramble short hairpin RNAs; MLL1sh: MLL1 short hairpin RNAs (knockdown); GelMA: porous gelatin methacryloyl hydrogel; ZIF-8: Zn(NO3)2 nanoparticles; AAV-5HRE-bFGF: hypoxia-response element (HRE) used to mediate human bFGF with adeno-associated virus (AAV).

**Table 2 cells-13-00817-t002:** Literature directly comparing the effects of dental-derived stem cells versus other mesenchymal stem cells on CNS secondary injury sequalae attenuation.

SCI/Non-SCI	Refer-ence	Stem Cell Types	Study Details	Secondary Injury Target Investigated	Superior DSC Activity Compared to Other MSCs
SCI	[107]	SHED vs. BMSCs	Rat SCI contusion model; cell free CM or SHED IS engraftment; in vitro analysis	Neuroinflammation; Angiogenesis; Apoptosis	CM functional recovery; spinal cord M2 gene expression; in vitro CM M2 macrophage induction; VEGF secretion; neuroprotective and anti-apoptotic factor release
SCI	[87]	hDPSCs, SHED vs. hBMSCs	Rat SCI transection model; SHED IS engraftment; in vitro analysis	Apoptosis/Neuro-protection	Neurotrophin expression; functional recovery; in vitro neurite extension
Non-SCI	[125]	hDPSCs vs. hPDLSCs, hBMSCs, hAMSCs	Mouse palatal mucosa injury model; stem cell injection	Structural events; Neuroinflammation	DPSC tissue regeneration; anti-inflammatory macrophage polarization
Non-SCI	[126]	hDPSCs vs. hAMSCs, hUMSCs	Mouse osteoporosis model; tail vein engraftment	Inflammation	Immunoregulatory potential of T-cell and macrophage anti-inflammatory polarization
Non-SCI	[127]	SHED vs. hBMSCs	Mouse allergic rhinitis model; IV engraftment	Inflammation	Reduced serum IgE and IgG1 levels; decreased inflammatory cytokines in spleen; modulation of T cells
Non-SCI	[128]	SHED vs. hBMSCs	Mouse systemic lupus erythematosus model; tail vein engraftment	Inflammation	Increased Treg cells to modulate inflammation
Non-SCI	[129]	hDPSCs vs. hBMSCs	Rat stroke model; hDPSC IV engraftment; in vitro ischemia analysis	Angiogenesis	IV engraftment efficacy; angiogenesis; in vitro neuroprotection; CM in vitro capillary formation
Non-SCI	[124]	SHED	Mouse Alzheimer’s disease model; SHED CM intranasal administration	Oxidative stress; Neuroinflammation; Neuroprotection/Anti-apoptosis	3-NT reduction; in vivo anti-inflammatory environment induction; neurotrophin release
Non-SCI	[130]	Murine DPSCs vs. BMSCs	In vitro and in vivo naïve mouse; tibialis anterior muscle injection	Angiogenesis	In vitro vessel formation; VEGF expression; in vivo vessel formation
Non-SCI in vitro	[90]	hDPSCs vs. hBMSCs	In vitro trigeminal and dorsal root ganglia microfluidic assay	Apoptosis/Neuro-protection	Neurotrophin expression; in vitro neuronal culture axon growth
Non-SCI in vitro	[123]	hDPSCs vs. hBMSCs vs. hAMSCs	In vitro axotomized rat RGC analysis	Neuroprotection/Neuritogenesis	RGC survival vs. hAMSCs; RGC neurite extension; neurotrophin expression; VEGF expression
Non-SCI in vitro	[131]	hDPSCs vs. hBMSCs	In vitro neurodegeneration analysis	Migration	Migration to neurodegenerative hippocampal neurons in vitro; expression of homing factors
Non-SCI in vitro	[88]	hDPSCs, hDFSCs, hSCAP vs. hBMSCs	In vitro neural differentiation analysis	Neural differentiation	Neural marker expression; CM induced neural differentiation of pre-neuroblastic cell line
Non-SCI in vitro	[132]	hDPSCs vs. hBMSCs	In vitro ischemia analysis	Oxidative stress	Ischemia-induced astrocyte death reduction by cells and CM

SHED: stem cells from human exfoliated deciduous teeth; BMSCs: bone marrow mesenchymal stem cells; IS: intra-spinal; CM: conditioned media; hDPSCs: human dental pulp stem cells; IV: intravenous; hAMSCs: adipose-derived mesenchymal stem cells; hUMSCs: umbilical cord-derived mesenchymal stem cells; RGC: retinal ganglion cells; hDFSCs: human dental follicle stem cells; hSCAP: human stem cells from apical papilla.

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
