# Peer review of "Molars to Medicine: A Focused Review on the Pre-Clinical Investigation and Treatment of Secondary Degeneration following Spinal Cord Injury Using Dental Stem Cells"

_cells, 2024, doi:10.3390/cells13100817_

Round 1

Reviewer 1 Report

Comments and Suggestions for Authors

“Molars to Medicine: A focused review on the pre-clinical investigation and treatment of secondary degeneration following spinal cord injury using dental stem cells” by Sandra Jenkner et al is timely review discussing possibility to use dental stem cells (DSCs) as one of the treatments for spinal cord injury (SCI). In my opinion, the review is not comprehensive enough and lacks a depth in some parts of the narrative.

Please find my comments and suggestions below.

Each sub-chapter should be either significantly expanded, or amended with the references to most recent comprehensive reviews on the subject of the sub-chapter. In the current state almost all sub-chapters are too short and lack a lot of relevant information/details.

“MSC have the capacity to differentiate into chondrocytes, myofibers, osteoblasts or osteocytes as well as neurons” - the ability of MSC cells to differentiate to neurons is questionable and a number of studies argue that MSC do not differentiate to functional neurons. Please discuss and provide corresponding references.

“Immune cells secrete high levels of pro-inflammatory interleukin (IL)-12, IL-23, IL-1b, TNF-a, IL-6, and low levels of anti-inflammatory IL-10”. What types of immune cells?

The authors mention apoptosis, but it's not the only types of the cell death caused by SCI (there are many!). Please discuss in detail or refer to other reviews on this subject (https://www.ncbi.nlm.nih.gov/pmc/articles/PMC7941236/ or others). There are many types of apoptosis as well. Please discuss.

Later in the text, the authors mention necrotic cell death, but only briefly (“ Acute changes <>, resulting in necrotic death of cells>).

Speaking about olygodendrocytes, how does myelin debris contribute to the tissue injury post-SCI?

“Numerous therapeutics have been studied in preclinical animal models” - what type of anymal models? What animals? Zebrafish, mice, rats, primates? Please specify when describing each case.

According to the authors of the review “The most common therapies are pharmacological (drugs or trophic factors) and cell or cell-derived therapies in nature [51]”, but the reference 51 is a review about animal models of SCI.

“Stem cell niches in the ependymal region surrounding the central canal have been postulated as a potential source of reparative endogenous cells [55]”-in what species?

Next, pharmacological management in acute spinal cord injury is comprehensively described in several recent works and not limited to methylprednisolone (see https://www.ncbi.nlm.nih.gov/pmc/articles/PMC10070428/, https://journals.lww.com/nrronline/fulltext/2023/01000/Molecular_approaches_for_spinal_cord_injury.3.aspx and many others)

It would be beneficial if authors listed all/most of the clinical trials involving DSCs and MSCs for SCI therapy, comparing their outcomes or preliminary results, to justify their statement that DSCs are superior as a stem cell therapy of SCI.

Figure 3. Please list the predominant neurotrophins secreted by DSCs.

How does the inflammed tissue microenvironment (as well as pharmacological therapies for SCI) affect the secretory milieu of DSCs?

“Data for intravenous administration show that only approximately 1-2% of circulating cells engraft into the spinal cord” - was the therapeutic effect observed in such a case?

It would be helpful (during the next rounds of the review) it the authors used an MDPI article template numbered lines.

Author Response

Please find Word document attached with point-by-point response to Reviewer 1.

Reviewer 2 Report

Comments and Suggestions for Authors

This review article describes the therapeutic potential of dental stem cells (dental pulp, dental follicle, apical papilla, etc.). The figures are practical and clear. The order of explanation is also practical; therefore, this is a basically nice review.  

In my opinion, the number of review articles cited in the references is slightly higher. The review of the reviews is one too many. The original article should be cited as much as possible, such as ref #70, through the text. The review of the reviews caused the bias, such as BMSC.

1.         The references #59-65 are likely all review articles published around 2010, and the materials are the bonemorrow MSC (BMSC) reported around 2000. As you know, a large number of reports on BMSC transplantation have been published around 2000, highlighting their multipotency. Their abilities were recently denied because there were less or no direct evidence of engraftment.  However, BMSC transplantation therapy has already been approved and has been used as the regenerative medicine ordered. This should be sought out, perhaps socially.

2.         The reports using human DPSC are particular important as the pre-clinical study in the Table 1. Thus, they should be separated from others.

3.         In vitro results are less important in regenerative medicine; therefore, it should be focused on in vivo results in Table 2 to clarify their effects.  

4.         Similarly, in the text, data from in vitro and in vivo studies are described as a mixture, which causes confusion. In my view, this is a weakness of this article. Certainly, the authors usually stated 'in vitro' at the end of sentences. However, readers always need to identify whether the experiment is in vitro or in vivo, whereas the story itself is interesting. It seems that sometimes the differentiation capacities are mixed in the same sentences. Basically, as the pre-clinical therapeutic meaning, treatment as equivalent in vitro and in vivo is not agreed.  

5.         Consequently, the authors should be identified in vitro and in vivo, and the human DSC and others. This should be need to the readers for easy understanding of the points of this review.   

6.         Additionally, the word mesenchymal stem cell (MSC) is in the broad sense of the term. Thus, I always want to ask that what kind of MSC. I know the authors used this word as 'bone morrow' MSC, but it is easier and more effective for the readers to use 'BMSC' through the text.  

Author Response

Please find Word document attached with point-by-point response to Reviewer 2.

Round 2

Reviewer 1 Report

Comments and Suggestions for Authors

The authors addressed all my previous comments and the manuscripts' quality has improved significantly. In its current form the review might be instrumental for whose working in a field of SCI/stem cell therapy.

Author Response

We thank Reviewer 1 for dedicating their time for the review our manuscript.

Reviewer 2 Report

Comments and Suggestions for Authors

The authors adequately responded to my comments.

Author Response

We thank Reviewer 2 for dedicating their time for the review our manuscript.